# Intelligence against complexity: Machine learning for nonuniform temperature-field measurements through laser absorption

**Ruiyuan Kang**[1], **Dimitrios C. Kyritsis**[1,2], **Panos Liatsis**[3]*

**1** Department of Mechanical Engineering, Khalifa University, Abu Dhabi, UAE, **2** Research and Innovation Center on $CO_2$ and Hydrogen, Khalifa University, Abu Dhabi, UAE, **3** Department of Electrical Engineering and Computer Science, Khalifa University, Abu Dhabi, UAE

* panos.liatsis@ku.ac.ae

**Data Availability Statement:** All related code and data are available on Github at https://github.com/RalphKang/nonuniformity-effect-on-LAS--temperature-measurement. Alternatively, one can

## Abstract

The effect of spatial nonuniformity of the temperature distribution was examined on the capability of machine-learning algorithms to provide accurate temperature prediction based on Laser Absorption Spectroscopy. First, sixteen machine learning models were trained as surrogate models of conventional physical methods to measure temperature from uniform temperature distributions (uniform-profile spectra). The best three of them, Gaussian Process Regression (GPR), VGG13, and Boosted Random Forest (BRF) were shown to work excellently on uniform profiles but their performance degraded tremendously on nonuniform-profile spectra. This indicated that directly using uniform-profile-targeted methods to nonuniform profiles was improper. However, after retraining models on nonuniform-profile data, the models of GPR and VGG13, which utilized all features of the spectra, not only showed good accuracy and sensitivity to spectral twins, but also showed excellent generalization performance on spectra of increased nonuniformity, which demonstrated that the negative effects of nonuniformity on temperature measurement could be overcome. In contrast, BRF, which utilized partial features, did not have good generalization performance, which implied the nonuniformity level had impact on regional features of spectra. By reducing the data dimensionality through T-SNE and LDA, the visualizations of the data in two-dimensional feature spaces demonstrated that two datasets of substantially different levels of non-uniformity shared very closely similar distributions in terms of both spectral appearance and spectrum-temperature mapping. Notably, datasets from uniform and nonuniform temperature distributions clustered in two different areas of the 2D spaces of the t-SNE and LDA features with very few samples overlapping.

## Introduction

Laser absorption Spectroscopy (LAS) is a fundamental temperature measurement tool, which is extensively used in combustion diagnostics [1], emission detection and quantification [2, 3]. Its relevance has been strengthened recently because of the realization that absorption by

check the release of the repository through doi: 10.5281/zenodo.7303621. Moreover, as a complement to the code, we uploaded the dataset and pretrained models on Kaggle, available through https://www.kaggle.com/datasets/ralphkang/laser-absorption-spectra-of-co2. In both the readme.md of the repository and description of the dataset, all necessary information and descriptions were provided to allow replication of results/reuse of models.

**Funding:** DCK would like to acknowledge partial support by Khalifa University through grant RC2-2019-009. The funders had no role in study design, data collection and analysis, decision to publish, or preparation of the manuscript.

**Competing interests:** The authors have declared that no competing interests exist.

molecules in the terrestrial atmosphere plays a substantial role in terms of climate change [4]. This is a line-of-sight measurement that cannot yield spatially resolved information. Consequently, LAS application is often restricted to laminar flows which possess a priori known profiles along the light path. Based on an assumption of uniform temperature profile, two-color methods [5] and line reversal [6] have been developed and applied for average temperature measurement. Machine learning models, which can be regarded as surrogate models of aforementioned physics-based methods, have also recently emerged in the field [7, 8].

Turbulent flows, which possess nonuniform and intensely unsteady temperature and concentration profiles are almost ubiquitous in practical applications, such as combustion in gas turbines [9] and scramjets [10]. To deal with such cases, some methods based on two-color pyrometry have been proposed. A robust idea is to do multiple two color measurements with different line pairs [11]. The multiple measurements can be used in order to calculate average temperature and imply the profile nonuniformity. However, the different combinations of multiple two-color measurements can lead to varying average temperature or nonuniformity level. Goldenstein, et, al. [12] judiciously selected lines of water which presented signal strengths that were closely linear with temperature, so that the temperature measured by this two-color pyrometry, could be equaled to the average temperature along the light path. However, these lines can only be approximated to be linear temperature dependence in a limited temperature range.

In addition to two-color measurements, there are other methods which use more lines or even entire bands. Profile fitting [13, 14], an upgraded line reversal, reconstructs the parameters of a profile shape function rather than iterating on temperature but with the constraint that the prior knowledge of the profile shape must be known. The Temperature Binning Distribution (TBD) method [15] does not need this prior knowledge, it retrieves temperature distribution bins by solving algebraic equation sets constructed by intensities of multiple lines at configured temperature bins, and the corresponding line intensities extracted from the experimental spectrum. However, the bins cannot be too finely divided, otherwise, a huge equation set must be solved, which may even be singular without unique solution. The Temperature Distribution Function method [16] is the most recent one, which utilizes the transformation of the spectrum and profile shape functions, but still, it is mainly available for simple and regular profile shapes.

In any case, the development of these methods has extended the feasibility of acquiring accurate temperature information along nonuniform profiles. It is also worth noting that these techniques do not provide any insights as to how profile nonuniformity affects the inaccuracy of the average temperature measurement. Understanding of this effect is most important, since temperature and concentration nonuniformity can affect spectral response in several interesting ways, two cases of which are shown in Fig 1. The first is a case of one uniform profile and one nonuniform profile that have the same average temperature and concentration, but can lead to different spectral appearances, as shown in Fig 1(a). According to this spectral discrepancy, applying the methods targeting uniform profile to nonuniform ones would introduce inaccuracy. The second case is when spectra have very closely the same measured spectrum, but are in fact generated from totally different profiles with distinct average temperatures, as shown in the Fig 1(b). This phenomenon is referred to as spectral twins. A question then rises as to whether spectra of twins can be distinguishable and estimated properly. Without enough understanding and consideration of the effect of nonuniformity on temperature measurement, which includes resolving the two situations that were considered above, it is difficult to provide accurate temperature measurements in non-uniform profiles.

Ideally, one would need to formulate an analytical model, which would illustrate how temperature measurement inaccuracy depends on level of nonuniformity, and, of course, the

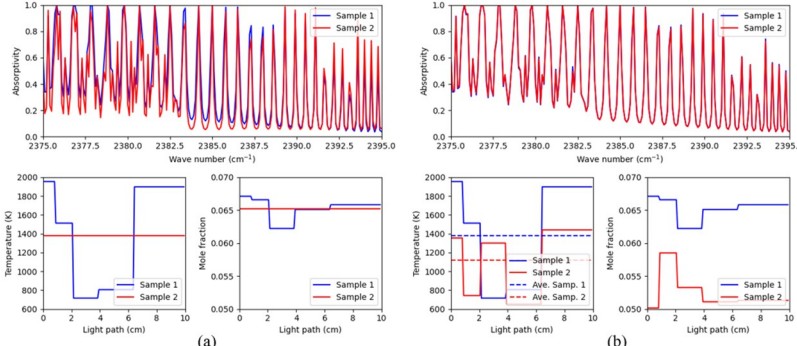

**Fig 1. Two cases which occur in profile nonuniformity.** In each figure, the top, bottom left and bottom right subplots represent the spectral profile, the temperature profile and the concentration profile, respectively. (a) The samples of uniform and nonuniform profiles have same average temperature and concentration but different spectral appearances. (b) Two samples have visually similar appearances but totally different temperature/concentration profiles and consequently different average temperatures.

specific methodology of measurement. However, it is unrealistically difficult to construct a specific analytical model of error propagation according to the mechanism of every method, something that has been illustrated very clearly in [12], where the analysis of nonuniformity effect on the simplest two-color method was determined by the complex coupling between temperature and concentration.

Rather than formulating (perhaps impossible) analytical models, we propose to use machine learning models, in order to use artificial intelligence to explore natural complexity. The logic behind is that, although different physics-based temperature measurement methods can operate on several physical principles, their data utilization fashion can be simply divided into two types, i.e., using few lines (regional features), such as two colors; or using multiple lines/bands (global features), such as profile fitting. Therefore, by training machine learning models with different data-utilization algorithms, we can summarize various physical models into several machine-learning-based surrogate models, and approach the optimal performance of each data-utilization algorithm. Through analyzing the performance of several such machine learning models, the nonuniformity effect can be quantified. Moreover, although machine learning models are in principle physics-agnostic, it is also true that they can be linked to different types of physical methods [17]. By analyzing the performance of various machine learning models in the particular physical problem, we could introduce the machine learning algorithms to the physical models that they represent in order to acquire physical insight about the temperature nonuniformity.

In this study, first, machine-learning-based models are trained to measure temperature from uniform-profile spectra. Then, we show that the application of the best three of the sixteen tested models, namely, Gaussian Process Regression (GPR) [18], VGG-13 [19] and Boosted Random Forest (BRF) [20], to nonuniform-profile spectra introduces significant measure error and poor capability of spectral twin distinguishment. In order to address this issue, these three models are retrained on spectra that were acquired from non-uniform profiles and applied to spectra of different nonuniformities. It is shown that nonuniformities can be captured with acceptable accuracy by models which utilize global features. In order to understand the precise nature of the issues that non-uniformity causes, we used T-distributed Stochastic Neighbor Embedding (t-SNE) [21] and Linear Discriminant Analysis (LDA) [22], which allow representation of the data in spaces of low dimensionality.

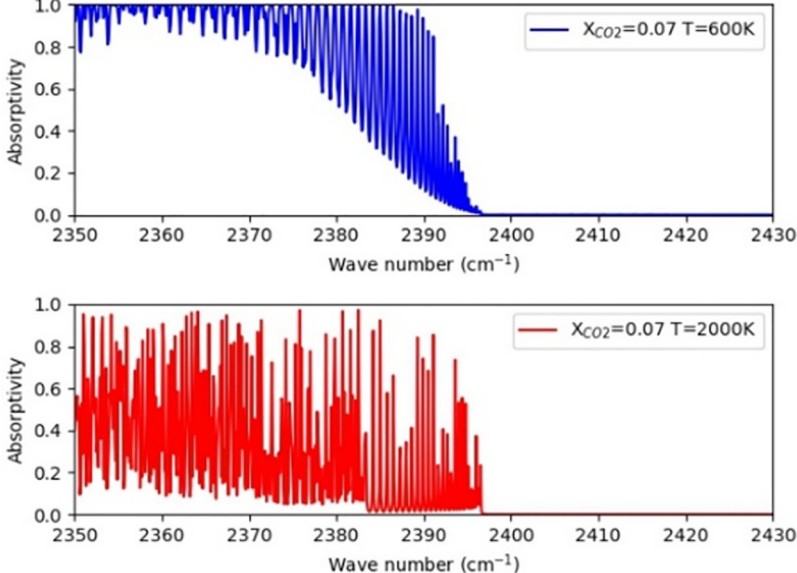

**Fig 2. The boundary cases of spectra at temperatures of 600 K (top) and 2000 K (bottom).**

## Spectral data preparation

### Spectral data generation

CO2 absorption spectra were synthesized using the HITRAN Application Programming Interface (HAPI) [23] and HITEMP 2010 databases [24], where, the substance, light path length, temperature, and mole fraction (concentration) were used as inputs in order to generate a spectrum.

Irrespectively of whether spectra are generated for the uniform or nonuniform profiles, the substance type, total light path length, temperature range and concentration range were controlled to be identical so that only the effect of nonuniformity determined the form of the generated spectra. The light-path length was set to 10 cm; the temperature was assigned from a typical combustion temperature range of 600-2000 K; and the wavelength band chosen is 2375-2395 $cm^{-1}$, because the line intensity in this range is sensitive to temperature changes, as shown in Fig 2. As for the mole fraction range, we chose it to be in the range of 0.05-0.07, which was deliberately selected in order to generate effective spectral twins.

In order to generate nonuniform profiles, we divided the light path into five sections, and randomly set the temperature and mole fraction in each section from the ranges aforementioned. In order to generate effective spectral twins, we had to avoid generating two types of trivially identical spectra samples, i.e., samples from different profiles with the same average temperature, concentration, and spectrum. One such case is two profiles that have the same temperature and mole fraction distributions, but just the ordering of the path sections differs when the light path is uniformly divided. This can be seen from the Beer-Lambert law (Eq 1):

$$\alpha_\nu = 1 - exp\left[-\sum_{i=1}^{5} S_\nu(T_i)\phi(T_i, P)\frac{P}{kT_i}x_i\Delta L\right] \tag{1}$$

Where, $\alpha$, $T$, $S$, $\phi$, $P$, $x$, and $\Delta L$ are absorptivity, temperature, line intensity, line shape function,

pressure, mole fraction, and section length, respectively. The subscripts v and i indicate frequency and section index respectively. Since the summation operation does not convey sequential spatial information, changing the order of the sections will not be reflected in the spectra. An example is shown in Fig 3(a), the profile of sample 2 is the profile of sample 1 with the order of first two sections changed. The spectra corresponding to these different profiles are still exactly same. In order to avoid such intrinsic identical samples, we divide the light path into a geometric sequence with the ratio of 1.4. Thus, the relative length of the sections is [1, 1.4, 1.96, 2.74, 5.38]. With such nonuniform light path division, the section exchange cannot lead to same spectra, as expressed by Eq 1.

However, this configuration is not sufficient to suppress a second type of trivially identical spectra, which have equivalent aggregated sections. In order to see how this can happen, consider Fig 3(b). With the exception of first two sections, the temperature and concentration profiles of the two samples are the same. Since the first two sections have equal temperature, and the average mole fraction in the aggregated section comprising the first two sections is equal to 0.05, the spectra generated will be identical. The trick to tackle this condition is to constrain the maximum ratio of the mole fraction between sections to be smaller than 1.4, the ratio of geometric sequence of section length. This is why the mole fraction in each section was selected between 0.05-0.07 (not including the value of 0.07).

After these precautions and in order not to generate trivially identical spectra, 10,000 samples of spectra were generated for uniform profiles and non-uniform profiles that comprised five sections of step-wise changing values of both mole fraction and temperature.

## Spectral twin selection

In order to assess the inaccuracy caused by spectral twins, we first selected spectral twins from the samples we generated. An algorithm for such purpose was developed and its principle is as follows. For a given spectrum, its twin counterpart should satisfy two requirements: (1) be similar to the given spectra, which is judged by Mean Square Error (MSE) between spectra lines smaller than a similarity index criterion $\epsilon$ and (2) that among all candidates satisfy the first requirement, it has the largest temperature difference from the given spectrum.

Here, we selected three levels of similarity index criterion, i.e., $\epsilon = 10^{-3}$, $10^{-4}$ and $10^{-5}$, in order to gradually diminish the MSE between twin spectra. Correspondingly, three datasets of spectral twins, were generated.

Table 1 summarizes the statistics of spectrum and spectral twins for these three datasets. When looking at the sample level, we observe that the average temperatures do not reach the

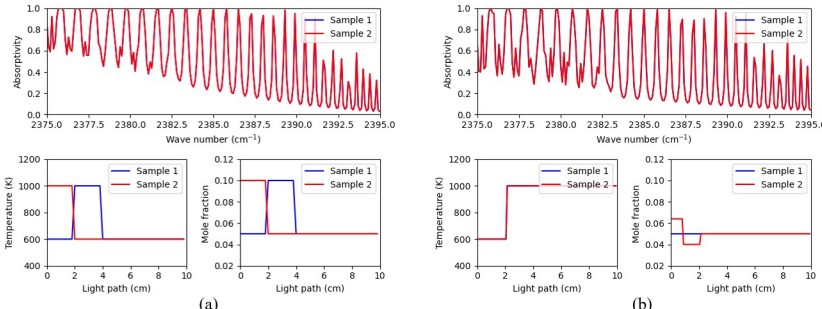

**Fig 3. Examples of two types of trivially identical spectra samples.** In each figure, the top, bottom left and bottom right subplots represent the spectral profile, the temperature profile and the concentration profile, respectively. (a) Case 1: Spectra generated from the same temperature/concentration distribution but different ordering of the segments. (b) Case 2: Profiles of two spectra with equivalent aggregated sections.

**Table 1. Statistics of datasets of spectral twins.**

| Similarity level | | | $10^{-3}$ | $10^{-4}$ | $10^{-5}$ |
|---|---|---|---|---|---|
| Sample level | Min (K) | | 648.28 | 648.28 | 741.26 |
| | Max (K) | | 1919.5 | 1919.5 | 1844.4 |
| | mean (K) | | 1301.8 | 1301.8 | 1309.8 |
| | range (K) | | 1271.2 | 1271.2 | 1103.2 |
| | Distribution | | Irwin-Hall | | |
| | Twin percentage | | 100% | 99.86% | 86.61% |
| Twin level | AE between twins (K) | Median | 231.9 | 142.4 | 69.5 |
| | | Max | 380.1 | 289.2 | 216.9 |
| | | Min | 63.2 | 0.9 | 6.99E-03 |
| | | 3rd quartile | 256.8 | 165.8 | 92.1 |
| | | Skewness | 0.174 | 0.139 | 0.25 |
| | RMSE between twins (K) | | 237 | 148.4 | 78 |
| | R between twins | | 0.46 | 0.75 | 0.92 |

configured boundaries, i.e., 600 and 2000 K. This is because in order to reach the boundary values of the temperature range, all five sections must have the boundary value, however, since the temperature was set independently in each section, reaching five boundary temperatures is a small probability event. Because temperature and concentration are set randomly, the average temperature of the nonuniform-profile data follows the Irwin–Hall distribution [25] for the distribution of the sum of n = 5 variables from multiple independent uniform distributions. Moreover, the stricter the criterion, the narrower the average temperature range that we can estimate. This is because when the similarity index criterion, i.e., MSE requirement, is stricter, it is harder for samples to find corresponding counterparts that will make up spectral twins. This is reflected in the decreasing percentage of twins detected. The spectra losing their twins mainly concentrate in both tails of the distribution. This is because they have fewer

**Table 2. Machine learning model performance on uniform-profile spectra.**

| Algorithm | Kernel/Architecture | RMSE (K) | R | Absolute Error (K) | | |
|---|---|---|---|---|---|---|
| | | | | Max | Mean | Median |
| SVR | RBF | 71.9 | 0.99 | 140.4 | 62.4 | 63.6 |
| | Poly | 64.5 | 0.99 | 141 | 53.89 | 49.9 |
| | Linear | 62.7 | 0.99 | 134.76 | 51.91 | 47.2 |
| GPR | Constant | 2.00E-03 | 1 | 3.20E-02 | 4.90E-05 | 4.90E-05 |
| | RBF | 4.20E-03 | 1 | 3.90E-02 | 2.60E-03 | 1.70E-03 |
| Ridge | Linear | 3.9 | 1 | 18.2 | 2.4 | 3 |
| | Poly | 12.8 | 1 | 51.1 | 8.4 | 4.8 |
| | RBF | 15.7 | 1 | 56.3 | 10.8 | 6.9 |
| XGBoost | BRF | 2.8 | 1 | 14.1 | 2.1 | 1.7 |
| NN | MLP | 2.7 | 1 | 20 | 1.5 | 0.9 |
| | VGG11 | 0.5 | 1 | 1.5 | 0.4 | 0.4 |
| | VGG13 | 0.3 | 1 | 1.5 | 0.2 | 0.2 |
| | Squeeze Net | 1.3 | 1 | 16.8 | 0.8 | 0.6 |
| | Xceptron | 0.4 | 1 | 5.8 | 0.3 | 0.2 |
| | Resnet18 | 0.5 | 1 | 7.8 | 0.3 | 0.3 |
| | Resnet34 | 1 | 1 | 29.3 | 0.3 | 0.2 |

candidates that can be spectral twins, thus causing the shrinkage of temperature range of datasets of spectral twins.

When examining the twin levels from Table 1, we see that the temperature difference between the reference spectra and their twin counterparts is decreasing while their correlations are increasing as the criteria are tightened, in terms of the changes in Absolute Error (AE), Root Mean Square Error (RMSE) and Pearson correlation coefficient (R). It is notable, however that, even at a similarity level of $10^{-5}$ when the highest correlation between spectral twins is achieved, the maximum temperature difference between twins can still reach up to 216.9 K, with a median RMSE and AE of about 70 K, which reflects the fact that significant temperature difference still exists between the twins under these conditions. This points to the importance of correct detection of the twins in the training algorithms that will follow.

## Quantifying the nonuniformity effect on temperature measurement

### Machine learning model training

Machine-learning models were trained as surrogate models of conventional physical methods to predict the average temperature from uniform-profile spectra. Specifically, 10,000 uniform-profile samples generated as described above were divided into training, validation, and test sets, in the numbers of 7000, 1500, 1500 samples, respectively. In total, we trained 16 machine learning models to predict temperatures from spectra of uniform profiles, as shown in Table 2.

The models can be categorized in five classes according to the learning process, i.e., Support Vector Regression (SVR) [26], Gaussian Process Regression (GPR) [18], Ridge regression [27], Boosted Random Forest (BRF) [20] in XGBoost [28], and Neural Network (NN). In SVR, GPR, and Ridge regression, a number of kernels were evaluated, e.g., radial basis function (RBF), polynomial (Poly), linear and constant, which are available in Scikit Learn [29]. In the case of NN, in addition to the conventional Multi-Layer Perceptron (MLP) [30], a number of Convolutional Neural Network (CNN) architectures [31] were considered, i.e., VGG [19], ResNet [32], Squeeze Net [33] and Xceptron [34]. These CNN architectures were originally devised for image classification tasks, but we modified them to be suitable for 1D spectrum data and regression tasks, by changing to 1D convolutional layers and removing the operations of batch normalization and dropout. More details about the machine learning algorithms can be found in the references cited.

The specific machine learning methods were chosen based on a consideration of their fundamental way of utilizing data. For instance, SVR uses support vectors, i.e., a tiny subset of the training dataset, in order to construct a hyperplane; while GPR is a non-parametric algorithm, which uses all features of all the samples of the training set to construct a correlation matrix as the reference for the test samples. Ridge regression also uses the information of all samples, since it is in fact a type of regularized least squares regression. BRF is an ensemble learning method, which is constructed by multiple tree models [35], but each one only uses partial features of subsets of training samples. As for NN, except the MLP which uses the raw features of the samples, all CNN architectures learn new features from the raw ones using the features of the entire training set. In summary, SVR and BRF utilize regional information, but the former utilizes partial samples, while the latter utilizes partial features, similar to a two-color measurement. The remaining three algorithms, GPR, Ridge and NN utilize global information, all features of all samples, which carries some similarity to the line-reversal method of classical spectroscopy.

A variety of metrics were used in order to assess model performance, including RMSE, R, and the maximum, mean and median of the AE. By examining the performance of the models in Table 2, we observe that using different kernels or variants of a machine learning algorithm

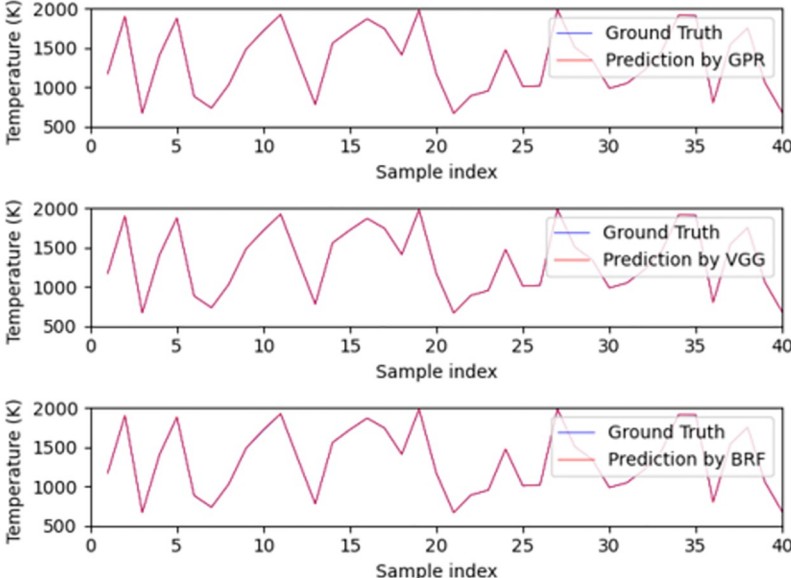

**Fig 4. The performance of the best three models on the test set for the prediction of uniformly distributed temperature.** The top, middle and bottom subplots are the predictions of GPR, VGG and BRF respectively.

does not have a substantial impact on performance. Moreover, we can see that the top performing method in terms of median absolute error is GPR, followed by NN, then BRF, Ridge regression, and finally SVR. In what follows, we choose the top three models, i.e., GPR with constant kernel (called as GPR in what follows), BRF and VGG13 (called as VGG in what follows), as the representatives of these surrogate models of physical methods. At the same time, these selections also represent three different ways of utilizing data as discussed above. From the demonstrations of Fig 4, it is observed that all these three models provide accurate predictions of temperature from uniform-profile spectra without noticeable differences.

## Generalization performance of models on nonuniform-profile data

We then applied GPR, VGG and BRF which were shown to be the top-performing algorithms of estimation of uniformly distributed temperature to spectra of nonuniform profiles comprising of five sections of randomly chosen temperature and mole fraction as explained above. The performance of these models is summarized in Table 3. By comparing Tables 2 to 3, it is evident that the performance of these models has dramatically degraded in the case of spectra of nonuniform profiles. It is worth noting that the maximum AE reaches up to 942 K, with its lowest value being at 615 K. The total temperature range is 2000-600 = 1400 K, which means that the maximum AE is at least 44% of the entire temperature range. When examining the

**Table 3. Model generalization on spectra of nonuniform profiles.**

| Model | RMSE (K) | R | Absolute Error (K) | | |
|---|---|---|---|---|---|
| | | | Max | Mean | Median |
| GPR | 310 | 0.74 | 942 | 267 | 244 |
| VGG | 216 | 0.86 | 684 | 188 | 172 |
| BRF | 206 | 0.89 | 615 | 182 | 171 |

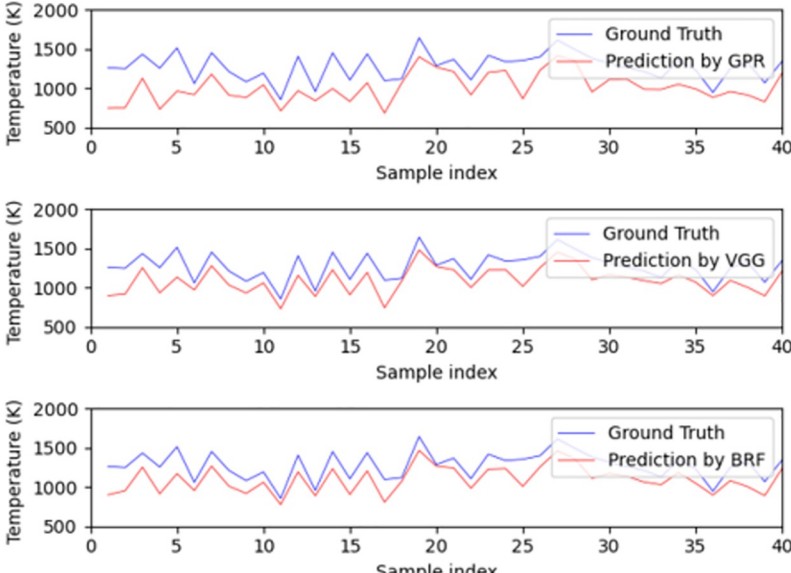

**Fig 5. Model generalization performance for the prediction of average temperature of non-uniform-profile spectra.** The top, middle and bottom subplots are the predictions of GPR, VGG and BRF, respectively.

RMSE, mean AE and Median AE metrics, they are at least one order of magnitude higher than those of the spectra of uniform profiles, which is always above 10% of the temperature range of the available data. These quantitative results indicate that directly using these models for uniform profiles will cause significantly inaccurate measurements of temperature based on the spectra of nonuniform profiles.

Although the models do not estimate the temperature accurately on nonuniform-profile data, they do capture trends in temperature correctly, as shown in Fig 5. This observation suggests that the nonuniform-profile data have some common properties with uniform-profile data. However, these two types of data also have substantial differences and as a result, the models performing well in predicting temperatures uniformly distributed in space cannot perform well in temperature estimations from non-uniform data.

Meanwhile, a very interesting observation is the performance rank of these three models on nonuniform-profile data, which is in fact in the inverse order of that of uniform-profile data, i.e., BRF > VGG > GPR. This may be due to two reasons. First, one model may overfit the uniform-profile data and underfit the nonuniform-profile data. Second, the machine learning method that uses partial features in an ensemble configuration is likely to be more robust than VGG which extracts new features from raw features. Clearly, GPR uses all the original features in the samples and is thus more sensitive to changes in the data, and thus demonstrates the least robust performance.

### Generalized performance of models on spectral twins

Because of the importance of the correct detection of spectral twins that was pointed out in section 2.2., this was also examined for the case of non-uniform profiles. The metrics of Relative Error (RE), R, Percentage of consistency ($P_{CST}$), and Percentage of Closeness ($P_{CLS}$) were

used to assess the performance of models on spectral twins. The definition of RE is as follows.

$$RE = \frac{|\widehat{T}_k - \widehat{T}_j|}{|T_k - T_j|} \tag{2}$$

where $T$ and $\widehat{T}$ are the ground truth and the prediction of average temperature, respectively; the subscripts k and j represent the reference spectrum and its counterpart for making up spectral twins, respectively.

In the best case, models should sense all individuals possessing the higher average temperature in the twin pairs and thus provide a corresponding higher temperature estimation. We defined $P_{CST}$ as the proportion in all estimations of spectral twins that satisfy this condition:

$$P_{CST} = \frac{c_1}{s} \tag{3}$$

$$c_1 = count\left(\frac{\widehat{T_{k,m}} - \widehat{T_{j,m}}}{T_{k,m} - T_{j,m}} > 0\right), m \in [1, s] \tag{4}$$

Where $s$ is the total number of spectral twins, and $m$ denotes the spectral twin index, which ranges from 1 to $s$. $c_1$ computes the total number of spectral-twin estimations which satisfy $\frac{\widehat{T_{k,m}} - \widehat{T_{j,m}}}{T_{k,m} - T_{j,m}} > 0$.

Additionally, the estimation of an individual of the spectral twin should be closer to the actual temperature of this individual rather than that of its counterpart in the twins, which can be expressed as that $\frac{|\widehat{T_{k,m}} - T_{k,m}|}{|\widehat{T_{k,m}} - T_{j,m}|} < 1$. Accordingly, we defined $P_{CLS}$ as shown in Eqs (5) and (6), which calculate the proportion of such cases among all pairs of spectral twins. Under the ideal condition, $P_{CLS}$ should also have a value of 1.

$$P_{CLS} = \frac{c_2}{s} \tag{5}$$

$$c_2 = count\left(\frac{|\widehat{T_{k,m}} - T_{k,m}|}{|\widehat{T_{k,m}} - T_{j,m}|} < 1\right), m \in [1, s] \tag{6}$$

The performance of the models is summarized in Table 4. The distribution of RE is highly right-skewed, as indicated by its large positive skewness; therefore, the median and 3rd quartile of RE are more statistically meaningful than the mean. In evaluating the results of Table 4, it is probably best to start with the "median" row, which provides a measure of the per-cent difference in average temperature that the several algorithms can "sense" as twins for a given

**Table 4. Performance of models on spectral twins.**

| Model | | GPR | | | VGG | | | BRF | | |
|---|---|---|---|---|---|---|---|---|---|---|
| Similarity criterion | | $10^{-3}$ | $10^{-4}$ | $10^{-5}$ | $10^{-3}$ | $10^{-4}$ | $10^{-5}$ | $10^{-3}$ | $10^{-4}$ | $10^{-5}$ |
| RE | Median | 0.32 | 0.22 | 0.14 | 0.17 | 0.13 | 0.1 | 0.14 | 0.1 | 0.08 |
| | 3rd quartile | 0.52 | 0.35 | 0.24 | 0.29 | 0.23 | 0.18 | 0.25 | 0.18 | 0.16 |
| | Skewness | 1.05 | 67.26 | 93.7 | 1.62 | 63.8 | 89.03 | 1.82 | 39.76 | 53.9 |
| R between estimations of twins | | 0.911 | 0.984 | 0.998 | 0.967 | 0.992 | 0.999 | 0.974 | 0.994 | 0.999 |
| $P_{CST}$ | | 0.2 | 0.21 | 0.3 | 0.59 | 0.57 | 0.56 | 0.7 | 0.64 | 0.59 |
| $P_{CLS}$ | | 0.58 | 0.54 | 0.53 | 0.58 | 0.54 | 0.53 | 0.58 | 0.54 | 0.53 |

similarity criterion. In this context, the larger this number, the better the performance of the algorithm in terms of twin detention, because then even twins with sizeable temperature differences can be detected. It is observed that the best result is realized by GPR on the dataset at $\varepsilon = 10^{-3}$, however even there, half of the estimations of these models are only able to capture twins with temperature differences 32% or smaller. The stricter similarity criterion causes even weaker ability of the models to distinguish twins with sizeable temperature differences, for instance, at the level of $= 10^{-5}$, the medians of RE dive to 0.14, 0.10 and 0.08 for GPR, VGG and BRF, respectively. Similar degrading performance trends are also observed in the metrics of R, and $P_{CST}$, except in the case of $P_{CST}$ for GPR, which is increasing with a stricter criterion. $P_{CLS}$ is just above 0.5 on all conditions, so whether the estimation is closer or not to the reference spectrum is fairly random.

Considering their previous poor accuracy in predicting average temperature from nonuniform-profile spectra, it is unsurprising that these models have poor performance on distinguishing spectral twins for spectra from non-uniform temperature distributions. In fact, as we showed in section 2.2., the sensitivity of distinguishing spectral twins and accuracy of temperature measurement are tightly correlated.

In addition, we can see that different algorithms have their strength on different metrics. GPR reaches largest values of Median and 3rd quartile of RE, and smallest values of R, which demonstrates that it has the best performance in providing estimations that can distinguish between spectral twins, even when these have sizeable temperature differences whereas, BRF demonstrates a diminished capability in this regard. On the other hand, in terms of $P_{CST}$, the ranking is reversed. This is because GPR is a very sensitive algorithm, which can detect slight differences between spectra. However, whether the estimations are right or not is a more complex matter. BRF, on the other hand, being an ensemble method, is more robust in detecting the temperature magnitude relationship between twins.

In Fig 6, we plot the performance of GPR, VGG, and BRF on the first 40 twins with the smallest and largest temperature difference at the similarity levels of $10^{-3}$ and $10^{-5}$. The twins with the largest temperature difference often stay around the middle of the temperature range, i.e., 1300 K. The twins with the smallest temperature difference mainly lie near the boundaries of the temperature range. The reason behind this behavior is the same as the one responsible for the Irwin-Hall distribution of the average temperature that was discussed in Section 2.2. Specifically, a spectrum implying a moderate average temperature can have a temperature profile derived from either a section combination of consistently moderate temperatures or a combination of high and low temperature sections complementing each other. However, a spectrum implying an extremely low or high temperature can only have a combination of sections of consistently low or high temperature profile. Clearly, spectra implying moderate temperatures have a higher degree of freedom in profile composition and consequently are prone to be composed of spectral twins with large temperature differences. Therefore, the spectral twins with the smallest temperature difference are not as representative and meaningful as the ones with the largest temperature difference, since they are merely limited by the data generation method and thus cannot assess the performance of models.

What also stands out in Fig 6 is that the models often underestimate the average temperatures for both spectral twins with largest and smallest temperature differences. This trend is particularly clear, when the similarity level is constrained to the $10^{-5}$ level. In addition, although models are capable to sometimes capture (to a limited degree) the contrast between spectral twins at the similarity level of $10^{-3}$, this frail sensitivity is eliminated by a stricter similarity criterion. such demonstrations match the observations from Table 4. Moreover, we can see that GPR is more sensitive to spectral feature changes than the other two machine learning

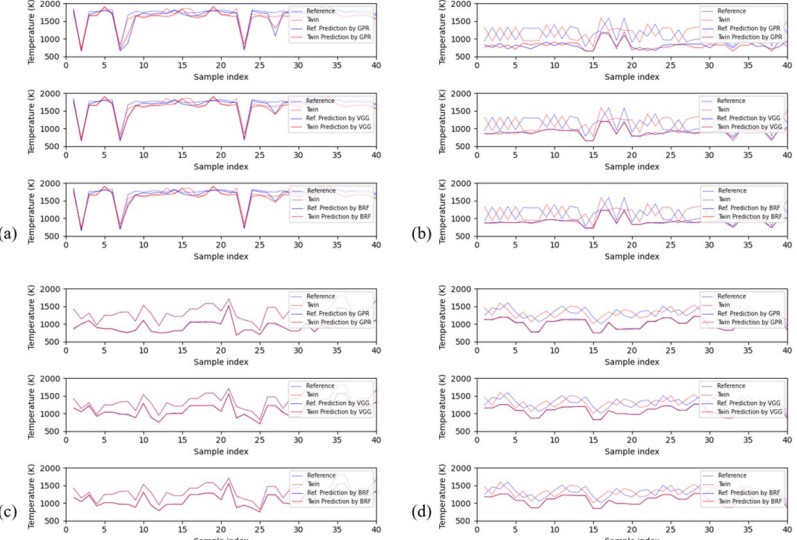

**Fig 6. Examples of performances of the models on spectral twins.** In each figure, the top, middle and bottom subplots are the predictions of GPR, VGG and BRF, respectively. (a) First 40 twins with the smallest temperature difference at the similarity level of $10^{-3}$. (b) First 40 twins with the biggest temperature difference at the similarity level of $10^{-3}$. (c) First 40 twins with the smallest temperature difference at the similarity level of $10^{-5}$. (d) First 40 twins with the biggest temperature difference at the similarity level of $10^{-5}$.

methods, since which is prone to provide more contrastive temperature estimations of spectral twins, while BRF is the least sensitive one.

## The feasibility of overcoming nonuniformity effect

### Dataset splitting for retraining models

The substantial negative impact of nonuniformity on the temperature measurement has been identified and quantified from the analysis of the previous section. The question arises as to whether this negative impact can be overcome. In order to address this issue, we retrained the models using nonuniform-profile spectra, with the expectation that the models can learn the spectrum-temperature mapping relationship of nonuniform-profile spectra and distinguish the differences between spectral twins.

The machine learning algorithms used were in the category of supervised learning, so that the labelling of spectra, i.e., average temperature, would navigate the learning direction of machine-learning models. As a result, the presence of spectral twins could lead to information leakage, in which the knowledge learnt from the samples in the training set could lead to incorrect predictions on the twins of these samples in the test set. To avoid this condition, we have to isolate the training and test sets.

For this reason, we applied a novel dataset splitting method, namely bidirectional isolation splitting, the principle of which is shown schematically in Fig 7. First, we duplicate the dataset of nonuniform-profile samples. The copies of the original spectra are represented as orange color boxes. Then, we split the samples into training set and test set with the partition of 70% and 30%, respectively. The dataset splitting boundary is represented by a black dash line in Fig 7. The links between spectral twins are represented by lines with arrows. To avoid information leakage, as long as a pair of spectral twins is composed of samples from both the training and the test sets (shown as red lines- irrespectively of whether a reference spectrum in training set

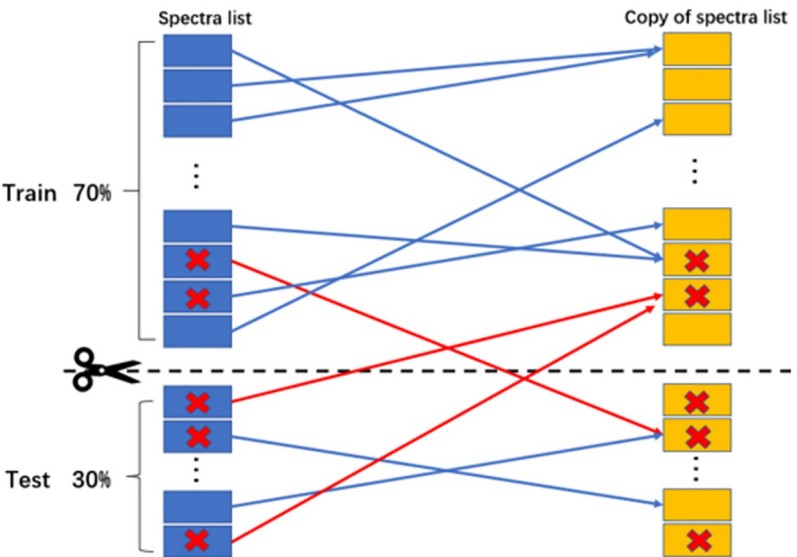

**Fig 7. Schematic representation of bidirectional isolation splitting.**

has its counterpart in test set or vice versa), these samples will be deleted from both training and test sets, i.e., bidirectional deleting will take place. Eventually, both training and test sets are self-contained and mutually exclusive, i.e., all samples in the training set only have their counterparts in the training set and so do the samples in test set. Of course, this kind of isolation is not thorough since one sample may have several similar counterparts, but bidirectional isolation can forbid the strongest effect from the most similar counterpart, which is termed twin. After this initial dataset splitting, the second bidirectional isolated splitting was only applied to the test set in order to divide it into two subsets with almost same number of samples, which were then used as the final test and validation sets.

The statistics of test sets from different similarity levels are shown in Table 5. Compared to Table 1, statistics in Table 5 displays diminished temperature ranges, which is reasonable since we have deleted some samples during dataset splitting. Other metrics such as mean and median, basically similar to the values in Table 1.

## Retraining and testing models on five-section-profile spectra

We retrained GPR, VGG and BRF, and summarized their performance in Table 6. In terms of sample level, all the models have decent performance on test sets, although it is not as good performance as on uniform-profile spectra shown in Table 2. However, the performance is substantially improved in terms of accuracy compared to the generalization performance of models trained on uniform-profile spectra, which was shown in Table 3. This implies that estimating accurate average temperatures from nonuniform-profile spectra is more difficult than estimating those from uniform-profile spectra, which is not surprising, but it also confirms that estimating average temperature from nonuniform-profile spectra is feasible, and the performance is acceptable.

As for the performance in terms of distinguishing between twins, which is also presented in Table 6, there is also substantial improvement in comparison to Table 4. This is again expected because of the close relation of the accuracy of the algorithms under consideration to the capability of the algorithms to distinguish between spectral twins.

**Table 5. Statistics of test sets for different similarity levels.**

| Similarity Level | | | $10^{-3}$ | $10^{-4}$ | $10^{-5}$ |
|---|---|---|---|---|---|
| Sample level | Min (K) | | 787.81 | 787.81 | 854.22 |
| | Max (K) | | 1816.11 | 1832.79 | 1761.22 |
| | Mean (K) | | 1314.47 | 1312.89 | 1315.57 |
| | Range (K) | | 1028.3 | 1044.98 | 907 |
| | Distribution | | Irwin-Hall | | |
| Twin Level | AE between twins | Median (K) | 229.357 | 141.87 | 70.01 |
| | | Max (K) | 347.759 | 261.5 | 170.6 |
| | | Min (K) | 112.42 | 11.51 | 5.22E-02 |
| | | $3^{rd}$ quartile (K) | 254.59 | 163.59 | 92.865 |
| | | Skewness | 0.329 | 0.182 | 0.2745 |
| | MSE (K) | | 236.149 | 147.95 | 79.166 |
| | R | | 0.422 | 0.7539 | 0.916 |

If we examine the effect of similarity levels, in addition to R and RMSE values, which are affected by the test set's similarity level, almost all other metrics in the twin level are also weakened, demonstrating that a stricter similarity criterion makes it more difficult to distinguish spectral twins. However, a higher similarity requirement reduces the estimation AE at the sample level, which is due to a stricter similarity level of samples forces models to figure out the details of spectral samples, as in adversarial learning [36].

In terms of median absolute error, the algorithms rank as GPR< VGG< BRF, which is the same ranking as the one observed in Table 2. This proves again that GPR, the algorithm that uses all features of the entire dataset, is most sensitive to slight differences between samples. On the contrary, BRF is the most insensitive one, since it cannot detect the subtle differences between spectral twins. As a result, the absolute error of BRF increases significantly when the similarity level changed to $10^{-5}$ from $10^{-4}$. This is also supported by the results of Fig 8, which show that the predictions of BRF do not always coincide with the ground truth of spectral twins.

**Table 6. Retrained model performance on test sets from different similarity level.**

| Metrics | | | GPR | | | VGG | | | BRF | | |
|---|---|---|---|---|---|---|---|---|---|---|---|
| Similarity level | | | $10^{-3}$ | $10^{-4}$ | $10^{-5}$ | $10^{-3}$ | $10^{-4}$ | $10^{-5}$ | $10^{-3}$ | $10^{-4}$ | $10^{-5}$ |
| Sample level | RMSE (K) | | 3.72 | 3.27 | 3.47 | 19.15 | 18.75 | 14.6 | 34.43 | 33.81 | 26.69 |
| | R | | 0.999 | 1 | 1 | 0.995 | 0.996 | 0.997 | 0.985 | 0.987 | 0.99 |
| | AE | Maximum | 42.22 | 25.14 | 42.72 | 68.34 | 69.05 | 55.21 | 113.4 | 114.84 | 109.95 |
| | | Median | 1.63 | 1.547 | 1.42 | 12.95 | 11.66 | 9.6 | 23.38 | 22.38 | 17.85 |
| Twin level | RE | Median | 0.999 | 0.998 | 0.998 | 0.881 | 0.833 | 0.811 | 0.669 | 0.5203 | 0.5236 |
| | | $3^{rd}$ quartile | 1.009 | 1.016 | 1.026 | 0.942 | 0.931 | 0.947 | 0.7713 | 0.6447 | 0.7732 |
| | | Skewness | -1.745 | 8.93 | 25.18 | -0.223 | -0.085 | 14.12 | -0.47 | 0.009 | 21.889 |
| | RMSE (K) | | 236.07 | 147.68 | 79.15 | 207.24 | 123.44 | 64.95 | 160.049 | 81.473 | 44.809 |
| | R | | 0.421 | 0.755 | 0.916 | 0.508 | 0.821 | 0.943 | 0.6804 | 0.9177 | 0.9726 |
| | $P_{CST}$ | | 1 | 1 | 0.989 | 1 | 1 | 0.972 | 1 | 0.993 | 0.911 |
| | $P_{CLS}$ | | 1 | 0.999 | 0.988 | 1 | 1 | 0.946 | 1 | 0.985 | 0.84 |

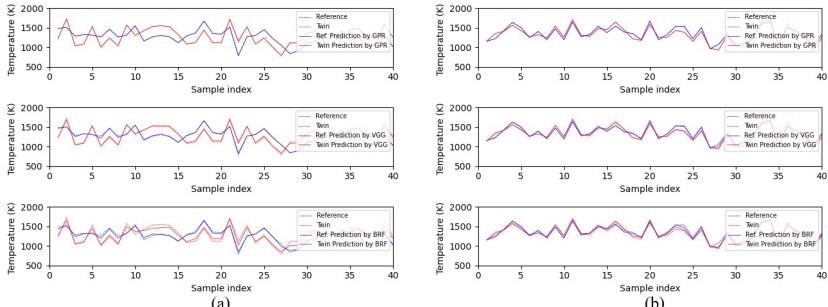

**Fig 8. Examples of performances of the retrained models on spectral twins.** In each figure, the top, middle and bottom subplots are the predictions of GPR, VGG and BRF, respectively. (a) First 40 twins with the biggest temperature difference at the similarity level of $10^{-3}$. (b) First 40 twins with the biggest temperature difference at the similarity level of $10^{-3}$.

## Generalization in different degrees of non-uniformity

### Increase nonuniformity

So far, we have examined the capabilities of the various algorithms with non-uniform profiles of five step-wise levels. The ability of the above models to achieve good generalization performance further than this artificial level of non-uniformity will determine whether the effect of nonuniformity can be truly overcome and the models trained can be regarded as potentially universal tools.

As a more complex profile compared to five-section profile, a ten-section profile was employed, the configurations of temperature, concentration, etc. were still set as before, but the light path is discretized uniformly rather than geometrically. 500 samples of this kind of profile were generated in order to test the already trained models. Samples from both five-section and ten-section profiles which have same average temperatures but different average concentrations are plotted in Fig 9. The spectra of these two types of profiles are apparently different, which is the combined result of different temperature nonuniformity and the concentration distribution.

We applied the models trained on five-section-profile data to ten-section profile spectra, and the performances of the models are summarized in Table 7. Comparing with the data of Table 6, the very encouraging (to an extent unanticipated) result is that the performance of both GPR and VGG is identical and not affected by the nonuniformity change, as also shown in Fig 10. On the other hand, the performance of BRF has significantly degraded. Also, consistently with its generalization performance on five-section-profile spectra (Fig 5), BRF underestimates the average temperatures of the spectra. The differences between model performance reveal that different nonuniformity levels cause regional differences in spectral appearance, but these regional differences are not strong enough to confuse the estimations of VGG and GPR. This is because GPR and VGG utilize all features of the distribution under consideration, while BRF utilizes multiple partial features. We could then hypothesize that the conventional physical methods which also utilize regional information, such as the two-color method, also suffer performance degradation when used without substantial consideration of the impact of nonuniformity.

### Decreased non-uniformity

Table 8 summarizes the generalization performance of the models on uniform-profile spectra, which are considered as a case of extremely low non-uniformity. The AE medians are relatively

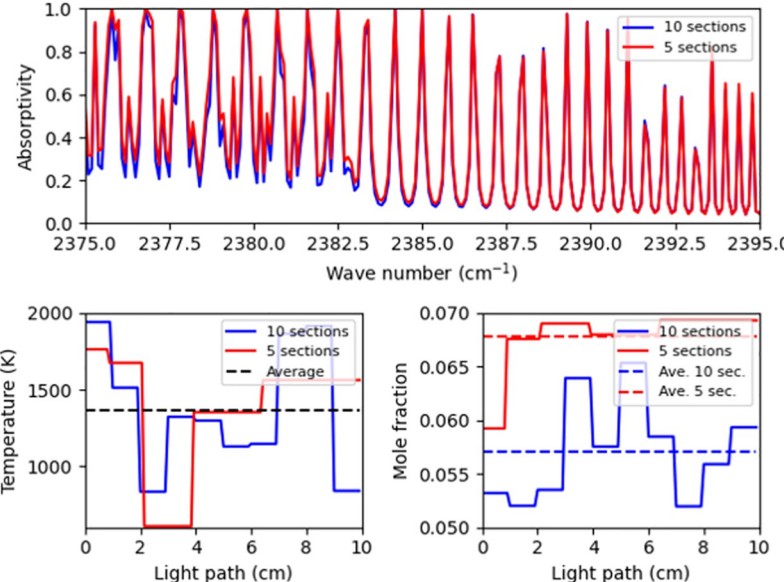

**Fig 9. A showcase of ten-section-profile spectrum, compared with five-section profile spectrum of same average temperature.** In the figure, the top, bottom left and bottom right subplots represent the spectral profile, the temperature profile and the concentration profile, respectively.

low except ones of BRF, but notably they are higher than their counterparts in Table 7. Because the median is not affected by outliers, i.e., extreme high estimation errors, therefore, the decent median proves that models of VGG and GPR still maintain excellent performance on uniform-profile spectra, although worse than their performance on ten-section-profile spectra. However, GPR and VGG acquire much higher AE maxima than those in Table 7, which can be higher than 1000 K and cause significant increase in mean AE, and RMSEs. The reason for this can be seen using Fig 11, which shows that VGG and GPR work pretty well on most samples, except for the cases of which the average temperatures are close to the boundary of the examined temperature range, e.g., 2000 K. The nonuniform-profile training set does not contain samples with such high-temperatures, as shown in Table 5. Therefore, the increased AE maxima, AE means, and RMSEs only prove that VGG and GPR are not good at extrapolation, although they demonstrate excellent interpolation performance. Fig 11 also shows that BRF

**Table 7. Generalization performance of the retrained models on ten-section-profile spectra.**

| Model | Twin level | R | RMSE (K) | Absolute Error (K) | | |
|---|---|---|---|---|---|---|
| | | | | Max | Mean | Median |
| GPR | $10^{-3}$ | 1 | 1.82 | 9.27 | 1.38 | 1.08 |
| | $10^{-4}$ | 1 | 1.93 | 7.61 | 1.47 | 1.2 |
| | $10^{-5}$ | 1 | 2.43 | 31.13 | 1.53 | 1.1 |
| VGG | $10^{-3}$ | 0.99 | 14.6 | 53.88 | 11.38 | 9.25 |
| | $10^{-4}$ | 0.99 | 14.87 | 50.96 | 11.56 | 9.87 |
| | $10^{-5}$ | 1 | 12.16 | 42.16 | 9.61 | 7.83 |
| BRF | $10^{-3}$ | 0.94 | 247.22 | 429.03 | 233.14 | 240.57 |
| | $10^{-4}$ | 0.92 | 183 | 406.808 | 142.01 | 92.98 |
| | $10^{-5}$ | 0.96 | 203.26 | 326.891 | 198.52 | 193.65 |

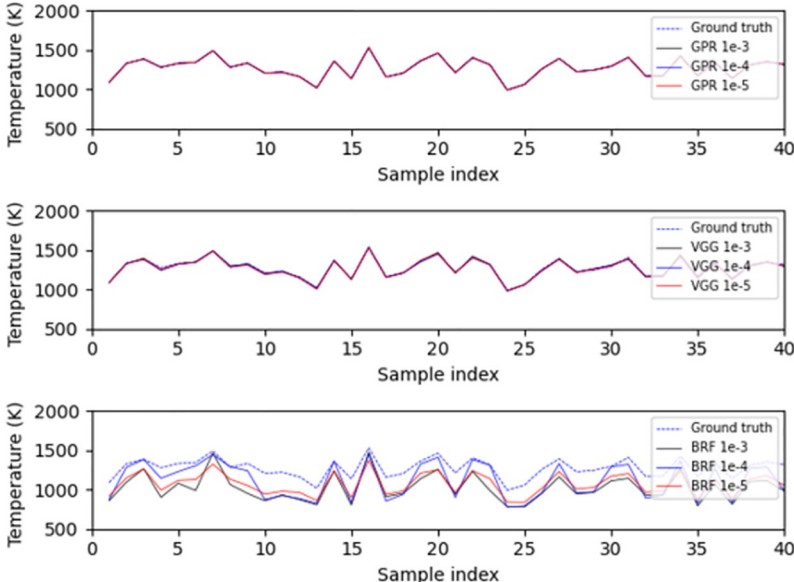

**Fig 10. Examples of generalization performance of the retrained models on ten-section-profile spectra.**

has relatively good performance in extrapolation but its prediction accuracy on most samples is unsatisfactory.

In closure, the generalization performance of GPR and VGG on both uniform or more nonuniform-profile samples proves that the negative effect of nonuniformity can be overcome regardless of the nonuniformity level of spectra. GPR and VGG, which use global features, are promising tools in order to perform spectral measurements of temperature from spatially non-uniform temperature distributions, with the condition that the prediction should be interpolation rather than extrapolation.

## Non-uniformity effect visualization

Although the examination of the performance of several algorithms have demonstrated that VGG and GPR can maintain decent performance with different degrees of nonuniformity, the reason behind this finding has not been shown in the previous analysis. In order to explore

**Table 8. Generalization performance of retrained models on uniform-profile spectra.**

| Model | Twin level | R | RMSE (K) | Absolute Error (K) | | |
|---|---|---|---|---|---|---|
| | | | | Max | Mean | Median |
| GPR | $10^{-3}$ | 0.93 | 153.13 | 1326.88 | 41.61 | 5.77 |
| | $10^{-4}$ | 0.96 | 112.99 | 1102.56 | 41.11 | 5.73 |
| | $10^{-5}$ | 0.94 | 133.87 | 1250.14 | 47.97 | 6.4 |
| VGG | $10^{-3}$ | 1 | 44.68 | 184.77 | 27 | 13 |
| | $10^{-4}$ | 0.99 | 49.56 | 317.73 | 27.92 | 15.2 |
| | $10^{-5}$ | 0.91 | 173.86 | 895.94 | 74.43 | 15.05 |
| BRF | $10^{-3}$ | 0.97 | 149.17 | 303.78 | 130.56 | 128.71 |
| | $10^{-4}$ | 0.97 | 138.2 | 292.69 | 115.2 | 106.3 |
| | $10^{-5}$ | 0.97 | 193.19 | 417.25 | 170.29 | 161.53 |

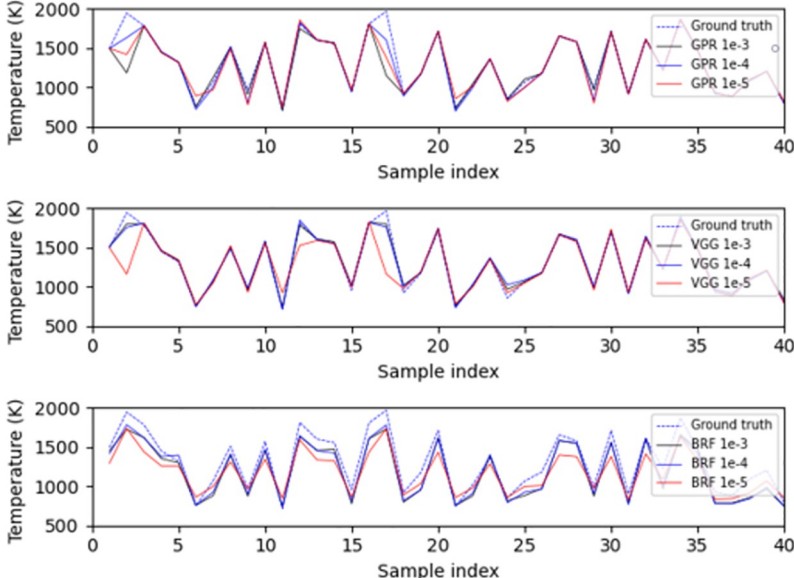

**Fig 11. Examples of generalization performance of the retrained models on uniform-profile spectra.**

such reasons, we used dimensionality reduction technologies to map the high dimensional spectra into two Dimensional (2D) features and then visualized the data in a two-dimensional space of features in order to obtain intuitive insights into the phenomenon.

Two dimensionality reduction technologies were attempted, they are t-distributed Stochastic Neighbor Embedding (t-SNE) [21] and Linear Discriminant Analysis (LDA) [22]. T-SNE is an unsupervised algorithm, which means no label information is provided during mapping high dimensional spectra to 2D features. T-SNE has the advantage of being able to maintain the relative distance relationship between samples after dimensionality reduction, thus allowing for the observation of structure of dataset distributions. LDA is a supervised algorithm. By utilizing the information of dataset belongings of spectral samples, it offers the possibility to amplify the differences between datasets of varying nonuniformity. More details of the algorithms can be found in the references cited therein.

T-SNE was first performed on randomly picked 500 samples from three datasets of different nonuniformities (i.e., uniform profiles, as well as profiles with five and ten different temperature levels). The high dimensional space that spectra sitting in was transformed to a 2D space, and the spectra were mapped to vectors comprised by two values. These 2D vectors (called as features in what follows) are then further plotted into Fig 12(a). Notably, the samples of five-section profiles and ten-section profiles are distributed in the same area in the 2D space of features, even though they have distinctive nonuniformity levels and light path divisions. The distribution of uniform-profile samples only has partial overlap with the distributions of nonuniform-profile samples, especially samples from five-section profiles.

This overlap between uniform- and nonuniform-profile datasets is almost eliminated in the results of the supervised LDA algorithm, as shown in Fig 12(b). The remained overlapping points are far from the cluster of nonuniform profile dataset but inside the cluster of uniform profile dataset. This phenomenon in fact implies a special case of nonuniform profile that uniform profile can be generated from individually random setting of discretized sections, despite it is a small probability event. Meanwhile, it is noted that, even when processing with the

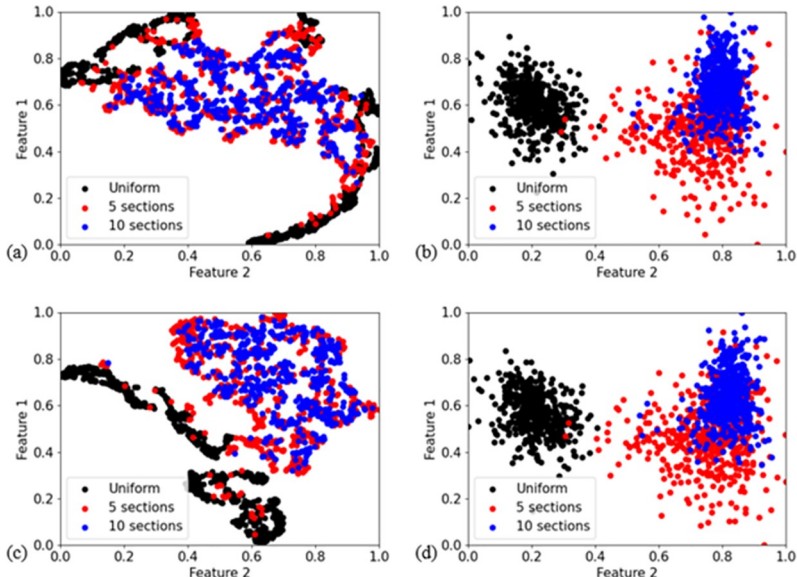

**Fig 12. Data visualization by t-SNE and LDA (a) t-SNE on spectra (b) LDA on spectra (c) t-SNE on spectra injected temperature (d) LDA on spectra injected temperature.**

strongly label-led LDA, five-section and ten-section-profile spectra datasets are still tightly matched. This phenomenon tells that these two nonuniform datasets share closely similar properties of in terms of spectral forms.

Apart from the correlation between datasets, it is more crucial to understand the correlation of spectra-temperature mapping relationship between datasets. Inspired by the idea of locating patch in Visual Transformer (ViT) [37], we injected temperature information into the spectral information by adding normalized temperature into every normalized line intensity of the spectra; therefore, the aggregated "spectra" contain the information of both raw spectra and their corresponding temperature information.

The updated feature maps of t-SNE and LDA are respectively corresponding to Fig 12(c) and 12(d). Comparing two feature maps generated by t-SNE, i.e., Fig 12(a) and 12(c), we see that the overlapping of uniform and nonuniform samples is reduced but still exists. The vanishing overlaps in fact correspond to the concept of spectral twins. However, the feature maps of the LDA, Fig 12(b) and 12(d), do not differ substantially, which is due to dataset belongings of samples have already been utilized in supervised LDA operation, the further detailed temperature information is somehow redundant. Moreover, it is noted that the datasets of five-section profile and ten-section profile remain in close vicinity in the reduced dimensionality space of either t-SNE or LDA. These observations indicate that two strongly nonuniform-profile datasets do not only have very similar spectra appearances but also have very similar spectra-temperature mapping relationship. This is the reason why VGG and GPR demonstrate good generalization performance on data sets of varying nonuniformity. In the meanwhile, the evident distinction between uniform- and non-uniform-profile datasets explains why models trained on uniform-profile datasets do not generalize well to non-uniform-profile datasets. Conversely, models trained on nonuniform profile datasets, on the other hand, can generalize decently to uniform profile datasets. This is because uniform profile can be generated with a small probability in a random setting of five sections, as discussed above, which is why a few

points of nonuniform-profile dataset leave their own dataset center but overlap the points in the cluster of uniform-profile datasets.

## Discussion

In this work, we provided quantitative results and associated discussions in terms of the effect of spatial non-uniformity with respect to the parameters of temperature and concentration distribution on the accuracy of the average temperature measurement estimate using laser absorption spectroscopy. The focus of the research was to explore the feasibility whether machine learning methods could provide reliable parameter estimates by overcoming this kind of effect.

Indeed, some earlier studies [11, 13, 15] provided the means to measure average temperature from absorption spectra subject to the presence of non-uniformity. However, they neither provided a systematic study of the effects of the non-uniformity, nor attempted to define its impact on the accuracy of the mean temperature measurements. For instance, in [12], there was an attempt to formally analyze and explain the mechanism of the effect of non-uniformities in temperature and distribution concentration. The outcomes of this work have merit, however, due to the complexities of the associated theoretical analysis in the simultaneous presence of non-uniformities in both temperature and concentration, the associated conclusions were not clear. In this research, we pursue a fundamentally different approach. The essence of the underlying concept is to use multiple machine-learning-based surrogate models, where each model was trained on uniform-profile data, to estimate the average temperature from nonuniform-profile data, and next, apply data analysis tools to quantify the effect of non-uniformity. Based on quantitative statistics, we confirmed that using temperature measurement tools for uniform profiles directly on spectra generated from non-uniform profiles introduces significant errors.

In addition, to the best of our knowledge, this is the first work, which attempts to overcome non-uniformity effects in laser absorption spectra by machine-learning-based solutions. The results demonstrate that some machine-learning-based solutions, based on global information, demonstrate robustness to changes in the magnitude and style of non-uniformity in temperature and concentration distributions, which could potentially solve the problem of measuring average temperature from laser absorption spectra, in the case of nonuniform profiles.

## Conclusion

In this study, we utilized machine learning to quantify and understand the negative effect of nonuniformity on LAS temperature measurement. The conclusions can be drawn as follows:

Although various machine-learning surrogate models of physical methods perform very satisfactory temperature prediction on spectra from areas of uniform temperature, they deteriorate significantly on nonuniform-profile spectra with huge measurement errors and negligible capability of distinguishing spectral twins. These results demonstrated that directly using uniform-profile-targeted methods to nonuniform-profile spectra is improper.

GPR and VGG retrained on five-section-profile spectra generalize well on spectra from different nonuniformities, whereas BRF cannot. This demonstrates the nonuniformity effect can be overcome by machine learning algorithms which use global features of the acquired spectra, something that is true for GPR and VGG, but not for BRF.

The good generalization performance of GPR and VGG was shown not to be affected by the degree of non-uniformity in the modelled spectra. This was rationalized by mapping the data in spaces of reduced dimensionality through t-SNE and LDA, which indicated that strongly nonuniform-profile spectra share the very similar properties in terms of both spectral

appearance and spectrum-temperature mapping in the context of the two features that t-SNA and LDA generated. In the low-dimensionality spaces and with the exception of very few overlaps, non-uniform-profile datasets were shown to possess distinct values of features from uniform-profile datasets, which is the reason why attempting generalization on spectra from non-uniform temperature distributions with algorithms that were trained on spectra from uniform profiles can lead to substantial inaccuracies.

## Acknowledgments

RK would like to acknowledge Dr. Aamna AlShehhi from Khalifa University for her course on advanced deep learning.

## Author Contributions

**Conceptualization:** Ruiyuan Kang, Dimitrios C. Kyritsis, Panos Liatsis.

**Data curation:** Ruiyuan Kang.

**Funding acquisition:** Dimitrios C. Kyritsis.

**Investigation:** Ruiyuan Kang.

**Methodology:** Dimitrios C. Kyritsis, Panos Liatsis.

**Project administration:** Dimitrios C. Kyritsis.

**Software:** Ruiyuan Kang.

**Supervision:** Dimitrios C. Kyritsis, Panos Liatsis.

**Validation:** Ruiyuan Kang.

**Visualization:** Ruiyuan Kang.

**Writing – original draft:** Ruiyuan Kang.

**Writing – review & editing:** Dimitrios C. Kyritsis, Panos Liatsis.

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
