## [Decision Letter · Decision Letter 0]

4 Nov 2022

PONE-D-22-24085Intelligence Against Complexity: Machine Learning for Nonuniform Temperature-field Measurements Through Laser AbsorptionPLOS ONE

Dear Dr. Liatsis,

Thank you for submitting your manuscript to PLOS ONE. After careful consideration, we feel that it has merit but does not fully meet PLOS ONE’s publication criteria as it currently stands. Therefore, we invite you to submit a revised version of the manuscript that addresses the points raised during the review process.

We look forward to receiving your revised manuscript.

Kind regards,

Jianguo Wang, PhD

Academic Editor

PLOS ONE

Journal Requirements:

 "DCK would like to acknowledge partial support by Khalifa University through grant RC2-2019-009"

   "DCK would like to acknowledge partial support by Khalifa University through grantRC2-2019-009"

  "DCK would like to acknowledge partial support by Khalifa University through grant RC2-2019-009"

   "NO authors have competing interests"

Reviewers' comments:

Reviewer's Responses to Questions

**Comments to the Author**

1. Is the manuscript technically sound, and do the data support the conclusions?

Reviewer #1: Yes

2. Has the statistical analysis been performed appropriately and rigorously? 

Reviewer #1: Yes

3. Have the authors made all data underlying the findings in their manuscript fully available?

Reviewer #1: Yes

4. Is the manuscript presented in an intelligible fashion and written in standard English?

Reviewer #1: Yes

5. Review Comments to the Author

Reviewer #1: This an outstanding paper, written with high quality style and good English language. The authors have compared between sixteen models for uniform-profile spectra and the best three models, i.e., GPR, VGG and BRF were further compared for non-uniform-profile data. The paper is well-documented and well strutured and presented in a clear manner. The results are deeply analyzed. I have no comment about the paper, except one minor.

1. I wonder if the authors can provide a shoot discussion, more precisely, a comparison of the present paper with other reported studies in the literature before the conclusion.

6. PLOS authors have the option to publish the peer review history of their article (what does this mean?). If published, this will include your full peer review and any attached files.

Reviewer #1: **Yes: **Pr. Salim Heddam

---

## [Author Response · Author response to Decision Letter 0]

9 Nov 2022

Response to the comments of the reviewer

R1.C1: This an outstanding paper, written with high quality style and good English language. The authors have compared between sixteen models for uniform-profile spectra and the best three models, i.e., GPR, VGG and BRF were further compared for non-uniform-profile data. The paper is well-documented and well strutured and presented in a clear manner. The results are deeply analyzed. I have no comment about the paper, except one minor 1. I wonder if the authors can provide a shoot discussion, more precisely, a comparison of the present paper with other reported studies in the literature before the conclusion.

Response: We are sincerely grateful to the reviewer for his encouraging comments and for recognizing the contributions of our work. We fully agree with the reviewer’s suggestion and in this respect, we incorporated a new section entitled “Discussion” as follows:

“In this work, we provided quantitative results and associated discussions in terms of the effect of spatial non-uniformity with respect to the parameters of temperature and concentration distribution on the accuracy of the average temperature measurement estimate using laser absorption spectroscopy. The focus of the research was to explore the feasibility whether machine learning methods could provide reliable parameter estimates by overcoming this kind of effect. Indeed, some earlier studies [11], [13], [15] provided the means to measure average temperature from absorption spectra subject to the presence of non-uniformity. However, they neither provided a systematic study of the effects of the non-uniformity, nor attempted to define its impact on the accuracy of the mean temperature measurements. For instance, in [12], there was an attempt to formally analyze and explain the mechanism of the effect of non-uniformities in temperature and distribution concentration. The outcomes of this work have merit, however, due to the complexities of the associated theoretical analysis in the simultaneous presence of non-uniformities in both temperature and concentration, the associated conclusions were not clear. In this research, we pursue a fundamentally different approach. The essence of the underlying concept is to use multiple machine-learning-based surrogate models, where each model was trained on uniform-profile data, to estimate the average temperature from nonuniform-profile data, and next, apply data analysis tools to quantify the effect of non-uniformity. Based on quantitative statistics, we confirmed that using temperature measurement tools for uniform profiles directly on spectra generated from non-uniform profiles introduces significant errors.

Lastly, to the best of our knowledge, this is the first work, which attempts to overcome non-uniformity effects in laser absorption spectra by machine-learning-based solutions. The results demonstrate that some machine-learning-based solutions, based on global information, demonstrate robustness to changes in the magnitude and style of non-uniformity in temperature and concentration distributions, which could potentially solve the problem of measuring average temperature from laser absorption spectra, in the case of nonuniform profiles.”

[11] Tudor Palaghita and Jerry Seitzman. “Control of Temperature Nonuniformity Based on Line-of-Sight Absorption”. In: 40th AIAA/ASME/SAE/ASEE Joint Propulsion Conference and Exhibit. 2004, p. 4163. doi: 10.2514/6.2004-4163.

[12] Christopher S. Goldenstein et al. “Two-Color Absorption Spectroscopy Strategy for Measuring the Column Density and Path Average Temperature of the Absorbing Species in Nonuniform Gases”. In: Applied Optics 52.33 (2013), pp. 7950–7962. doi: 10.1364/AO.52.007950.

[13] Xiang Liu, Jay B. Jeffries, and Ronald K. Hanson. “Measurement of Nonuniform Temperature Distributions Using Line-of-Sight Absorption Spectroscopy”. In: AIAA Journal 45.2 (2007), pp. 411–419. doi: 10.2514/1.26708.

[15] Scott T. Sanders et al. “Diode-Laser Absorption Sensor for Line-of-Sight Gas Temperature Distributions”. In: Applied Optics 40.24 (2001), pp. 4404–4404. doi: 10.1364/ao.40.004404.

---

## [Decision Letter · Decision Letter 1]

28 Nov 2022

Intelligence Against Complexity: Machine Learning for Nonuniform Temperature-field Measurements Through Laser Absorption

PONE-D-22-24085R1

Dear Dr. Liatsis,

We’re pleased to inform you that your manuscript has been judged scientifically suitable for publication and will be formally accepted for publication once it meets all outstanding technical requirements.

Kind regards,

Jianguo Wang, PhD

Academic Editor

PLOS ONE

Additional Editor Comments (optional):

Reviewers' comments:

Reviewer's Responses to Questions

**Comments to the Author**

1. If the authors have adequately addressed your comments raised in a previous round of review and you feel that this manuscript is now acceptable for publication, you may indicate that here to bypass the “Comments to the Author” section, enter your conflict of interest statement in the “Confidential to Editor” section, and submit your "Accept" recommendation.

Reviewer #1: All comments have been addressed

2. Is the manuscript technically sound, and do the data support the conclusions?

Reviewer #1: Yes

3. Has the statistical analysis been performed appropriately and rigorously? 

Reviewer #1: I Don't Know

4. Have the authors made all data underlying the findings in their manuscript fully available?

Reviewer #1: Yes

5. Is the manuscript presented in an intelligible fashion and written in standard English?

Reviewer #1: Yes

6. Review Comments to the Author

Reviewer #1: The authors have provided the necessary section (Discussion) and the paper is now ready for publication. No further revision is necessary.

7. PLOS authors have the option to publish the peer review history of their article (what does this mean?). If published, this will include your full peer review and any attached files.

Reviewer #1: **Yes: **Pr. Salim Heddam

---

## [Editor Report · Acceptance letter]

1 Dec 2022

PONE-D-22-24085R1 

Intelligence Against Complexity: Machine Learning for Nonuniform Temperature-field Measurements Through Laser Absorption 

Dear Dr. Liatsis:

I'm pleased to inform you that your manuscript has been deemed suitable for publication in PLOS ONE. Congratulations! Your manuscript is now with our production department. 

Kind regards, 

on behalf of

Dr. Jianguo Wang 

Academic Editor

PLOS ONE